# TSCH-Sim: Scaling Up Simulations of TSCH and 6TiSCH Networks

**DOI:** 10.3390/s20195663

**Published:** 2020-10-03

**Authors:** Atis Elsts

**Affiliations:** Institute of Electronics and Computer Science, 14 Dzerbenes St, LV-1006 Riga, Latvia; atis.elsts@edi.lv

**Keywords:** TSCH, 6TiSCH, IoT, low-power wireless, simulation

## Abstract

TSCH (Time-Slotted Channel Hopping) and 6TiSCH (IPv6 over the TSCH mode of IEEE 802.15.4e) low-power wireless networks are becoming prominent in the industrial Internet of Things (IoT) and other areas where high reliability is needed in conjunction with energy efficiency. Due to the complexity of IoT deployments, network simulations are typically used for pre-deployment design and validation. However, it is currently difficult and time-consuming to simulate large-scale IoT networks with thousands of nodes. This paper proposes TSCH-Sim: a new discrete event simulator for IEEE 802.15.4-2015 TSCH and 6TiSCH networks. The evaluation shows that simulation results obtained with TSCH-Sim show a good match with results from other simulators that are commonly used to investigate TSCH networks. At the same time, TSCH-Sim is faster than these alternatives at least by an order of magnitude, making it more practical to carry out simulations of large networks.

## 1. Introduction

The TSCH (Time-Slotted Channel Hopping) protocol [1] is suitable for wireless network applications where high reliability and predictability is desired in conjunction with low power consumption. These include applications in industrial IoT [2], smart homes for healthcare [3], smart buildings [4], agricultural monitoring [5], and other areas.

Simulation is a critical step in the workflow of designing and deploying such networks. It is also an actively used research tool. To this end, the 6TiSCH (IPv6 over the TSCH mode of IEEE 802.15.4e) working group [6] has developed the 6TiSCH simulator [7]—a discrete event simulator that implements the TSCH and 6TiSCH protocols. Another frequently used tool for TSCH network research and validation is the Cooja simulator [8], which allows use of the Contiki and Contiki-NG implementations of TSCH & 6TiSCH protocols in simulated environments. While highly useful, these simulators have limited scalability [7] for simulations of large networks and when many scenarios need to be investigated (Section 4). This problem is becoming more acute with each year, as IoT networks expand in popularity and in size, and TSCH starts to get applied to LoRa-based networks [9] where large deployments are commonplace.

This paper presents TSCH-Sim: a new discrete event simulator for TSCH and 6TiSCH networks, available at https://github.com/edi-riga/tsch-sim. TSCH-Sim is implemented in JavaScript with modules to facilitate extensions by users and integration with the Web. It supports the TSCH protocol as defined in the IEEE 802.15.4-2015 standard [1], as well as aspects of the emerging 6TiSCH standards, including integration of TSCH with IPv6 and the RPL (Routing Protocol for Low-Power and Lossy Networks) protocol [10]. It allows the selection of multiple schedulers including the 6TiSCH minimal schedule and Orchestra [11], routing protocols (currently RPL), network connectivity models, including the models available in 6TiSCH and Cooja, node mobility models, as well as full control over the low-level details of the simulations such as the channel hopping sequence, queue size, and duration of individual timeslots. TSCH-Sim automatically collects simulation statistics for each run, including end-to-end Packet Delivery Ratio (PDR), hop-by-hop Packet ACK Ratio (PAR), radio duty cycle (RDC) and charge consumption for each node. The model of charge consumption is validated with hardware measurements on a Texas Instruments CC2650 device. TSCH-Sim has an easy-to-use web interface and uses the JSON format for configuration files. The simulator features a built-in random number generator that allows conduct replicable experiments, and a configuration option for running the same experiment multiple times with different random seeds, executed concurrently in case of a multicore CPU. Most importantly, TSCH-Sim demonstrates an order of magnitude faster execution time for medium-sized and large networks compared with other simulators evaluated in this paper, mostly because of a high abstraction level, improved link processing, and reduced log file sizes. In particular, in sparse networks it demonstrates approximately linear dependence between simulation speed and the number of nodes.

The paper is structured as follows: Section 2 presents the background on the network protocols and overviews the existing simulation options. Section 3 briefly describes the main aspects of TSCH-Sim. Section 4 evaluates TSCH-Sim and quantitatively compares it with existing options, and Section 5 provides an application example for the simulator. Finally, Section 6 concludes the paper.

## 2. Background and Related Work

### 2.1. TSCH and 6TiSCH

The IEEE 802.15.4-2015 standard [1] introduces several new MAC modes of operation, including the Time-Slotted Channel Hopping (TSCH) protocol, a Time Division Multiple Access (TDMA) MAC layer using frequency diversity via pseudo-random channel hopping. TSCH is targeted towards applications that require high reliability and predictability. Because of channel hopping at the MAC layer, TSCH networks are relatively immune to narrow-band interference and fading. Time in TSCH networks (Figure 1) is divided in *timeslots*, and all communication takes place on scheduled *cells*. A collection of periodically repeating cells is called a *slotframe*, and a collection of such slotframes is called a *schedule*. A cell has two coordinates: time slot and channel offset. The former corresponds to the time since the start of the network as expressed by a Absolute Slot Number (ASN). The latter is a mapped to a physical channel via a pseudo-random network-wide channel hopping sequence (HS):channel=HS[(ASN+ChannelOffset)mod||HS||]

The IETF (Internet Engineering Task Force) 6TiSCH Working Group has produced a number of Internet Drafts and RFC that describe how to use the TSCH protocol in low-power IPv6 networks. They have published the 6TiSCH minimal schedule RFC [6] and are, at the time of writing this, working on the Internet Draft of the Minimal Scheduling Function [12] for distributed TSCH scheduling.

The two most commonly used simulators for these protocols in the research literature are Cooja and the 6TiSCH simulator.

### 2.2. Cooja Simulator

Cooja [8] is a simulator for the Contiki [13] operating system. It is written in Java and is extensible by users via a plugin mechanism. It is possible to run native C code in the Cooja simulator using Java Native Interface (JNI). In particular, the complete Contiki and Contiki-NG operating system can be executed in the Cooja simulator, thus ensuring a close match between the software used in simulations and the software used in real-world deployments. Furthermore, using the *mspsim* plugin, emulation of msp430-based hardware platforms is possible.

However, this high level of detail comes at a cost: the slow performance of the hardware-emulated devices makes them impractical for networks with 100 or more nodes. Even when the hardware emulation feature is not used, Cooja is only suitable for networks with up to a few hundreds of nodes. Please note that the original Cooja paper [8] claims linear scalability with the number of nodes (*N*). However, the experiments in that paper do not include any packet transmissions—a major oversight given that the real bottleneck of Cooja’s performance is its connectivity model. In fact, it was the realization that $O(N^{2})$ algorithms are highly embedded in Cooja’s core code that first motivated the development of TSCH-Sim.

### 2.3. 6TiSCH Simulator

The 6TiSCH simulator is developed by the IETF 6TiSCH working group and serves as an experimentation ground for the standards they are developing. This simulator does not aim to simulate execution at the level of C code or machine code; it is written in Python and simulates time with the granularity of individual TSCH timeslots. The simulator collects summary statistics and stores them after each run; visualization options are also available, both in the form of a web interface and PDF files generated by Python scripts included in the source code release.

As the 6TiSCH simulator is intended to be used for experimentation with standards and proposed standards, it is lacking any non-standard schedulers and routing protocols, and does not have architecture that would facilitate the addition of custom extensions. While the 6TiSCH simulator is significantly faster than Cooja due to its higher level of abstraction, its scalability is limited due to the architectural choices made by its authors.

### 2.4. Other Simulators

#### 2.4.1. OpenSim

OpenSim (https://openwsn.atlassian.net/wiki/spaces/OW/pages/13434892/OpenSim) is the simulator of the OpenWSN [14] library, which is the *de facto* standard TSCH and 6TiSCH implementation for embedded devices. OpenSim allows running of the OpenWSN firmware on emulated devices connected over a simulated network. The main goals of OpenSim are to facilitate experimentation with OpenWSN without a hardware device, and to allow the developers testing their code before running it on hardware. Research claims that its scalability is rather poor [7], since low-level details of the device are emulated. The connectivity models in OpenSim are very simple and partially share code with the 6TiSCH simulator, as the same authors are involved in both. Specifically, the OpenSim simulator supports the Pister-hack distance-based model, line model, and full mesh model.

OpenSim does not support trace-based simulations or explicitly configured links, therefore it is not possible to guarantee a fair experimental comparison between it and TSCH-Sim and Cooja, as the aim is to set up an identical physical network topology before starting the simulations.

#### 2.4.2. ns-3

The ns-3 simulator (https://www.nsnam.org/) is the third version of the *ns* (“network simulator”). ns-3 was initially released in 2008, and has grown to contain a large collection of network protocols, topology definition tools, propagation models and other features. The simulator is written in C++ and has a Python interface for scripting. While some research papers also report results using the ns-3 simulator, its official release does not include an implementation of TSCH or 6TiSCH.

There is an unofficial ns-3 implementation of TSCH (https://github.com/EIT-ICT-RICH/ns-3-dev-TSCH). However, the implementation does not include all aspects of TSCH: for example, there is no concept of joining the network and no TSCH protocol messages are implemented, including both Enhanced Beacon and Keep-alive messages. Support of any 6TiSCH aspects is missing as well: there is no RPL implementation in the version of ns-3 that includes the partial TSCH implementation. These missing features make it impossible to perform an experimental comparison between the ns-3 TSCH implementation and TSCH-Sim.

#### 2.4.3. OMNeT++

OMNeT++ (https://omnetpp.org/) is an extensible, modular, component-based C++ simulation library and framework, primarily for building network simulators. OMNeT++ features a large number of protocols, traffic models, and ability to describe the network topology with a custom language called NED. Wireless channel and radio models are available not in the core library itself, but via other tools and simulators that rely on the OMNeT++ platform. OMNeT++ does not include support for the TSCH protocol.

The strengths of OMNeT++ and ns-3 lie in their ability to simulate radio reception models in high detail, including state-of-the-art channel fading and interference models, other physical layer details such as antenna directionality, propagation delay, as well as having a large library of common Internet protocols and devices. However, the majority of these features are not relevant to creating scalable TSCH and 6TiSCH simulations. TSCH-Sim does not aim to replace ns-3 or OMNeT++ with their detailed physical layers, but provide a simple alternative for simulating one specific protocol.

## 3. TSCH-Sim Overview

### 3.1. Core Functionality

TSCH-Sim is implemented in JavaScript using classes and code separation in modules. This language is selected to make the simulator easy-to-extend by users, as well as to facilitate the integration with a web frontend. Table 1 shows a summary of TSCH-Sim core features and their comparison with other simulators.

The TSCH-Sim code is loosely structured in three layers (Figure 2). The highest of these is the *interface layer* with modules that keeps the global configuration, route web API calls, and perform logging. A web GUI frontend is available (Figure 3). The intermediate is the *network layer*, which implements the upper half of the core simulator functionality. Multiple link connectivity options are available, including models that estimate packet reception probability and RSSI (Received signal strength indication) based on the distance between nodes, models that emulate options available on Cooja and the 6TiSCH simulator, and a trace-based model. Similarly, line mobility and random-waypoint mobility models are available. Finally, the *device layer* implements the state of a network node, including the IEEE TSCH protocol state machine. This layer also features multiple routing and scheduling options, which can be selected via configuration. Besides RPL, a simple leaf-and-forwarder routing is available. In terms of schedulers, Orchestra, leaf-and-forwarder scheduler, and the 6TiSCH minimal scheduler are currently implemented.

The JavaScript language includes a random number generator (RNG). However, this RNG is not suitable for simulation purposes, as JavaScript does not allow seeding of its RNG from user code, making it impossible to conduct repeatable simulations. TSCH-Sim features a custom random number module that uses the multiply-with-carry algorithm. This RNG algorithm characterized by fast execution speed and ability to pass statistical tests of randomness very well [15]. However, the multiply-with-carry algorithm generates uniform random numbers, and for some aspects of simulation the Gaussian distribution is required: for example, RSSI on communication links is characterized by additive white Gaussian noise. The random number module computes the Box-Muller transform [16] to convert uniform random numbers to the Gaussian distribution.

The TSCH-Sim simulator does not impose any specific slotframe structure: this depends in the scheduler. By default, TSCH-Sim has the Orchestra scheduler enabled. This scheduler adds three slotframes: one for node-specific unicast cells, one for sending and receiving Enhanced Beacon messages, and one called “common default”, shared by all nodes in the network and used for broadcast messages, such as RPL DIO messages. See Duquennoy et al. [11] and Elsts et al. [17] for the details.

### 3.2. Modeling Packet Transmissions and Collisions

The discrete nature of TSCH slotframe simplifies packet transmission modeling as packets may only be sent at a specific point in each timeslot. In each TSCH timeslot, all nodes in the network are processed and a scheduling decision made for each node. These decisions are possible: (1) transit a packet; (2) listen for a packet; (3) sleep. Nodes that transmit their packets do that by going through their links, checking whether the potential receiver is listening for packets and is on the right channel. If these preconditions are true, the link model is queried for transmission success probability between the transmitter and receiver nodes, a random number between 0 and 1 generated to determine the success for the packet, and the packet added on the receiver node: either to potentially received packets if the transmission is successful, or to interfering packets if it is not.

If the receiver node only has a single potentially received packet and no interfering packets in a given timeslot, the one packet is simply received. If there is more than one packet, Algorithm 1 implements reception with the capture effect [18]. In the algorithm’s pseudo-code, functions dBm_to_mW() and mW_to_dBm() refer to conversion between RSSI, as commonly given in decibel-milliwatts, and transmission power in milliwatts. The goal is to determine whether any of the packets can still be received, which is the case if the strongest RSSI is more than the sum of the other RSSI minus the hardware-specific co-channel rejection threshold (by default −3 dBm in TSCH-Sim).
**Algorithm 1.** Packet reception with collisions.1:**function** ReceivePacket(potential_packets, interfering_packets)2:  best_packet← SelectByStrongestRSSI(potential_packets)3:  best_packet_signal_mW←dBm_to_mW(best_packet.rssi)4:  other_packet_signal_mW←05:  **for**
packet∈potential_packets
**do**6:   **if**
packet≠best_packet
**then**7:    other_packet_signal_mW←other_packet_signal_mW+dBm_to_mW(packet.rssi)8:   **end if**9:  **end for**10:  **for**
packet∈interfering_packets
**do**11:   other_packet_signal_mW←other_packet_signal_mW+dBm_to_mW(packet.rssi)12:  **end for**13:  **if**
mW_to_dBm(best_packet_signal_mW) >14:    mW_to_dBm(other_packet_signal_mW) −co_channel_rejection
**then**15:   **return**
best_packet16:  **else**17:   **return**
NULL18:  **end if**19:**end function**

### 3.3. Achieving High Performance

The TSCH-Sim connectivity model only considers packet reception possible if there is an existing link between the transmitter node and receiver node. The existence of these links is precalculated at the start of the simulation by the link model (Section 3.5) and these links are kept in a cache for fast access. Due to this optimization, the execution time of TSCH-Sim simulations does not grow quadratically with the number of nodes (*N*), but linearly with the number of links (*L*). This is important in particular for sparse networks where $L\ll N^{2}$, as this property enables scaling up the network size as long as the number of potential connections per node remains small.

Cooja and the 6TiSCH simulator use similar strategies; however, the details are different. Cooja also caches the links in a hash table, but updates the links from a particular device whenever its radio is turned on or off. In a heavily duty-cycled protocol such a TSCH this creates a large overhead. The 6TiSCH simulator uses a $N^{2}$ connectivity matrix to provide fast access for all connections. The drawback of this approach is that whenever a transmission is done, all O(N) listening nodes need to be processed to the ones able to hear the packet, instead of iterating through the list of links from the transmitter with $O(\frac{L}{N})$ expected size.

The second main performance optimization included in the TSCH-Sim is related to the collection of simulation results. The 6TiSCH simulator and Cooja use a two-step approach for result collection: during the simulation run, unstructured or loosely structured results are logged in an output file. After the simulation is completed, the file needs to be analyzed with a script to produce summary metrics. To reconstruct the main performance metrics (Section 4.2), heavy log output is required; this takes a lot of disk space, and a lot of the runtime is spent waiting for I/O. The 6TiSCH simulator in particular creates files larger than a gigabyte for the 300 node simulations (Section 4.3). TSCH-Sim is able to generate detailed log files as well; in fact, the main performance metrics (Section 4.2) were reconstructed from these log files for a fair comparison with the other simulators. However, the TSCH-Sim code itself can collect all the required statistics during a simulation run, and report it immediately after the simulation is completed. This allows fully disabling of the log output and achievement of faster execution time without losing essential information about the results.

### 3.4. User Workflow

The main task of the user is to prepare a configuration for simulation runs. The configuration can be prepared either using the web interface or by directly writing the values in a JSON file; see the Listing 1 for an example. Example files are available along the source code of the simulator (Section 3.7).

Listing 1: TSCH-Sim configuration file example.

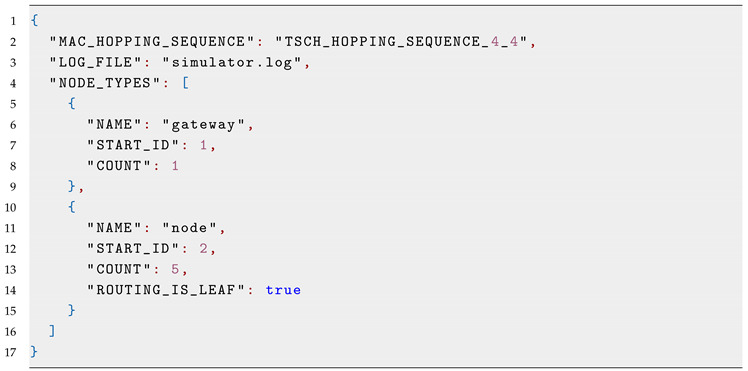



Sensible default values for all settings are provided, with the exception of the number of nodes and their types. The configuration file is only required to contain values that differ from these defaults. The following information can be specified in the configuration file:Simulation duration, simulation seed, the size of the simulated area;Log file name and logging levels, including module-specific levels;Link connectivity and node mobility models to use;Routing protocol and scheduler to use;Application-layer settings, such as the data packet size and generation period;Link model settings, such as the transmission range;Explicit link quality to use between specific nodes, if not covered by a distance-based link model;Mobility model settings, such as the speed of nodes;Routing protocol settings, such as the RPL packet periods, the lifetime of routes, whether to use the leaf mode, and whether to use RPL probing;Scheduler settings, such as the TSCH slotframe size and timeslot duration;MAC-layer settings, such as the number of retransmissions, queue size, TSCH Enhanced Beacon period, TSCH keep-alive packet period, and so on;Node types (see below).

The configuration key “NODE_TYPES” is expected to contain an array defining *node types* to be included in the simulation. Each node type must have the following parameters: its name and the number of nodes of this type. It can also contain the starting ID of the first node of this type, the data packet generation frequency and the destination ID for data packets, as well as all other configuration parameters whose values are different from the top-level configuration values or the default settings. Consider an example: a network with two types of devices: full-functionality devices capable of routing and forwarding, and reduced functionality devices capable of connecting as leaf nodes. The latter are going to have “ROUTING_IS_LEAF” set to true in the configuration of their node. In contrast, the former may leave this key unspecified, as its default value is false.

There is a constant packet rate model, configurable from the JSON and implemented the “PacketSource” class. The user can attach a packet source to a node or a group of nodes and configure parameters such as the application warm-up time, packet frequency, packet size, and packet destination. It allows simulation of data collection, data dissemination, and data query applications.

Once the configuration is ready, the user may start the simulation, either by pressing the “Start” button in the web interface, or by running the simulator from command line with the configuration file name as a command-line parameter. As the simulation is executed, the output from the logging module is printed both to the standard output and to a log file. At the end of the simulation, summary metrics are calculated and saved in a JSON file, both for each node and for the simulation as a whole. Unless otherwise specified, the results are stored in the results directory under the simulator’s installation path.

### 3.5. Link and Mobility Models

All links in TSCH-Sim are unidirectional. This allows the creation of simulations with asymmetric communication: for example, there may be a good link from node *A* to node *B*, perhaps because *A* has high transmission power, but a lossy or no link from *B* to *A*. Unless a link is explicitly configured, its quality and signal strength depends on the distance between the source and destination nodes.

#### 3.5.1. Explicitly Configured Links

For some radio links, their quality and other parameters may be fixed by the configuration. These are called “fixed” links in TSCH-Sim. The main options are link quality and the average RSSI. The link quality denotes the probability of a successful reception. There are two options for the format of this setting. First, it can be a number between 0 and 1.0, with the default value of 1.0. Second, it can be a dictionary with numerical keys corresponding to channel numbers in the TSCH channel hopping sequence, and a single value for each key, between 0 and 1. The latter allows the configuring of per-channel link qualities.

#### 3.5.2. Unit Disk Graph Model

The unit disk graph model is the default propagation model used in Cooja and in other simulators. The unit disk graph radio signal propagation model uses circle centered at the transmitting mode to simulate radio coverage. The reception success probability normally depends on the distance *d* between the transmitting and the receiving node, and is calculated using the formula $P\left(success\right)=1.0-d^{2}\cdot \left(1.0-UDGM\_RX\_SUCCESS\right)$, where UDGM_RX_SUCCESS is a configuration value. However, if UDGM_CONSTANT_LOSS configuration setting is enabled, the reception success probability is constant everywhere in the unit disk.

#### 3.5.3. Logistic Loss Model

The Logistic Loss radio signal propagation model uses circle centered at the transmitting mode to simulate radio coverage. The main role of Logistic Loss model is to serve as a more realistic replacement for the unit disk graph model.

The signal strength at the receiving node is determined using exponential free-space loss formula, on which Additive White Gaussian Noise (AWGN) is overlaid. The packet reception probability is modeled using the logistic function from the signal strength:(1)$$PRR\left(rssi\right)=\frac{1}{1+exp(-\left(rssi-rssi_{50\%}\right))},$$
where rssi is the transmit power minus the path loss. To model the path loss $PL_{dBm}\left(d\right)$ we use the log-distance path loss model [19]:(2)$$PL_{dBm}\left(d\right)=PL_{0}+10\cdot \alpha \cdot \log_{10}\frac{d}{d_{0}}+N\left(0,\sigma \right),$$
where by default the transmission range $d_{0}=200$ m, $PL_{0}=-100$ dBm [20], $rssi_{50\%}=-96$ dBm, α=3, and σ=3 [19].

#### 3.5.4. Pister-Hack Model

The Pister-hack model is the default propagation model used both in the 6TiSCH simulator and in OpenWSN OpenSim. The “hack” in the name refers to the property of the model is used to tune the RSSI levels to better match empirical results: namely the RSSI is randomly selected to be between the value predicted by the Friis model and to a value 40 dB below the Friis model.

#### 3.5.5. Trace-Based Simulation

Packet reception traces allow the input of real-world radio link dynamics in simulated network. This can significantly improve the fidelity of simulations to real-world testbed. However, trace-based simulation is still no replacement for real-world experiments, as short-term dynamics are typically missed by the trace preparation process. The TSCH-Sim simulator uses a trace file format from the 6TiSCH simulator called “K7”. Both plaintext files and compressed files (with gzip) are support. The latter are expected to have .gz extension.

#### 3.5.6. Mobility Support

Two mobility models are supported in TSCH-Sim:Random-waypoint mobility model: a node generates a random point in the mobility range and moves towards that point. Once the point is reached, the mode repeats the process.Line-based mobility model: a node moves along a single line segment. Once one end of the segment is reached, the node starts moving backwards, towards the other end, and so on.

Different mobility parameters can be enabled for different node types. For example, single simulation may combine static nodes, slow randomly moving nodes, fast randomly moving nodes, and nodes moving along multiple predefined lines.

### 3.6. Charge Consumption Model

When a simulation is executed in TSCH-Sim, the simulator keeps track of a number of statistics, including the number of slots used on each node. For the benefit of the user, at the end of the simulation, these slot-usage statistics are translated into charge consumption, which in turn can be used to estimate the battery lifetime on each node.

The charge consumption values used in the TSCH-Sim code are based on hardware measurements obtained using a RocketLogger device [21] on a Texas Instruments CC2650 [20] board. We measure the current consumption profiles of individual TSCH timeslots 100 times for each of the following: transmission of 50, 75, 100, and 123 byte sized packets (123 being the maximal MAC-layer payload) and reception of 50, 75, 100, and 123 byte sized packets, as well as idle listening slots. IEEE 802.15.4 standard encryption is enabled for all packets, and the CC2650 hardware acceleration for AES-128 is turned on.

Based on the results (Figure 4), we run linear regressions to estimate the charge consumption for packet sizes not directly measured. The regressions show very good fit, with $R^{2}=0.93$ for transmissions with ACK and $R^{2}>0.96$ for each of the three other experiments (Figure 5). The results allow a conclusion that this is a reasonably accurate method for estimating charge consumption in TSCH networks.

TSCH-Sim calculates the charge consumed by a given node by accounting all packets that it has sent and received, as well as all slots it has spent in idle listening and scanning the channel. Depending on whether a packet is received or transmitted, and whether it has an ACK or not, the simulator selects one of the previously determined regression equations. Then, the charge for the packet is calculated by applying the selected regression equation on the size of the packet. To calculate the total charge of idle listening, a hardware-validated constant value is used as the charge of each idle listening slot, multiplied by the number of such slots. All these charges are summed up and reported as the total charge consumed by the node, both in microcoulumbs and converted to milliampere-hours (mAh).

### 3.7. Examples

The simulator comes with a number of examples demonstrating specific network topologies and features of the simulator. It also includes a number of regression tests for all main simulation features. The following examples are included:Demos with star, mesh, line and grid networks, all under examplesDemo with a two-hop hierarchical network: examples/hierarchicalDemo with a large network with 1000 nodes: examples/large-networkDemo that demonstrates parallelization of multiple simulation runs: examples/multirunDemo that demonstrates control of a simulation with a user-defined script: examples/scriptingDemo that launches the web interface backend: examples/webDemo with comparative simulation runs and result visualization using matplotlib with Python: examples/result-visualization

## 4. Evaluation

### 4.1. Experimental Setup

All experiments described in the paper are run on a Lenovo X1 Carbon laptop with Intel i7-10710U CPU and 16 GB of RAM. The following software is used:OS for all simulators: Ubuntu 20.04.1 LTS, Linux kernel version 5.4.0-42-generic SMP x86_64.TSCH-Sim: Node.js v10.19.0 (Node.js v8 is the minimum required).Python 3.8.2 for the 6TiSCH simulator (Version used available at https://bit.ly/3hPpyg1).Cooja: Contiki-NG Docker image with OpenJDK Runtime Environment (version 1.8.0_222), Contiki-NG v4.5 (Version used available at https://bit.ly/351SZrO).

### 4.2. Simulation Fidelity

First, the TSCH-Sim simulator is compared with Cooja and the 6TiSCH simulator to validate the fidelity of its simulation results. Several connected multihop networks with 30 nodes are randomly generated. The links of these networks are hard-coded in the simulator configurations, using the TSCH-Sim configuration files, the DirectedGraphMedium plugin of Cooja, and the k7 trace files of the 6TiSCH simulator. A simple configuration supported by all three simulators is used: the 6TiSCH minimal schedule (with slotframe size 13) and the RPL routing protocol with the OF0 objective function, with the parameters from Table 2; these parameters are also used for all subsequent experiments described on this paper. The application packet size as given in Table 2 includes IP layer headers, but not MAC or PHY layer headers. To the author’s surprise, the 6TiSCH simulator implements a non-standard network joining algorithm; to minimize the discrepancy of the simulation results due the difference in joining times, for this evaluation we modified the 6TiSCH simulator by extending it with the standard IEEE 802.15.4 EB generation algorithm.

The performance of various metrics is evaluated on 10 randomly generated networks, with three different random seeds on each network. Figure 6 shows the quantitative comparison results; the distributions show aggregate results from the 3×10=30 simulations. Overall, the results support the hypothesis that TSCH-Sim approximates the results of Cooja and the 6TiSCH simulator with high fidelity. There is a good match between the three simulators with little systemic differences. For most of the metrics, the inter-simulator variance of that metric does not exceed the intra-simulator variance.

Figure 7 shows two routing tree topologies created in different simulation runs. While the topologies are not identical, they have high similarity. By counting the number of nodes with a different parents in Figure 7a,b, one can obtain a tree-similarity metric: tree edit distance. It is just 2 for in this case.

Figure 8 shows the histogram of the tree edit distances between the different simulation runs executed on the same underlying link topologies. The median distance between different TSCH-Sim runs is slightly larger than 5, demonstrating that RPL networks do not usually converge to the same routing tree. The median distance between TSCH-Sim and Cooja run is slightly smaller than that of TSCH-Sim itself, indicating that there are no systemic differences between the different simulators in this aspect. A *t*-test shows that the difference in the results is not statistically significant. The same goes for TSCH-Sim and the 6TiSCH simulator.

### 4.3. Simulation Speed and Scalability

#### 4.3.1. Comparison with Other Simulators

To compare TSCH-Sim speed with Cooja and the 6TiSCH simulator, we generate random connected multihop networks with up to 300 nodes per network and measure the speed it takes to execute an hour long simulation depending on the network size. The results show that TSCH-Sim is approximately an order of magnitude faster than the 6TiSCH simulator and approximately two orders of magnitude faster than Cooja. This speedup by a factor obviously translates to an increased amount of wall-clock time as the simulation duration increases. In Figure 9, simulation execution time can be measured depending on network size.

We note that the 6TiSCH simulator speed could be greatly improved by disabling or partially disabling logging. However, due to the fact that summary metrics in this simulator are generated indirectly, from log files of simulation runs, doing so would make the simulator impossible to compare in terms of fidelity, rendering the comparison unfair.

#### 4.3.2. Scalability in Hierarchical Networks

To further show the scalability of TSCH-Sim, we analyze the simulation performance in even larger networks. The focus of these simulations is on sparse networks where the number of actual links is much lower than the number of potential connections. Commonly used scheduling options such as Orchestra or the 6TiSCH minimal scheduler are not appropriate for networks with thousands of nodes. To work around this limitation, we select a simpler two-hop hierarchical topology instead of random multihop networks. Each network with this topology has a large number of leaf nodes $N_{L}$, much fewer forwarder nodes ($N_{F}\approx \sqrt{N_{L}}$), and a single gateway node. Each leaf node is only connected to a single forwarder, and each forwarder is connected to the gateway. A leaf-and-forwarder routing mechanism is used instead of RPL (both are available as options in the TSCH-Sim configuration), as well as a leaf-and-forwarder schedule. In this schedule, the gateway and forwarder nodes listen for potential packets in every slot, while leaf nodes only have two active slots per slotframe: one for broadcast packet exchange, and one for unicast data transmission to their associated forwarder nodes.

Networks with up to 10,000 nodes are simulated with these settings (Figure 10). It takes 11 min and 22 s to simulate an hour of the 10,000 node network operation, i.e., speedup of a factor of 5 compared to the wall-clock time. The expected slowdown when more nodes are added is slightly worse than linear, as can be seen from the regression line in the Figure 10. This suggests that even larger networks could be easily simulated.

#### 4.3.3. Impact of Mobility

TSCH-Sim keeps a precalculated cache of links that speed up packet transmission success rate calculations. The cache is periodically updated for links to and from mobile nodes (Section 3.3). Shorter update period leads to more accurate results, but reduces the simulation speed. The default setting of 10 s allows for maximally fast simulations that are only accurate if the nodes are moving slowly. Figure 11 shows that increasing the update frequency up to 10 times per second does not significantly slow the simulations down; however, update frequency of 100 times per second does slow the simulation down more than twice. Updating the links more frequently than is currently not possible: as the default TSCH slot size is 10 ms, and packet transmissions can only happen at specific points in the slot, the accuracy gains would be small.

## 5. Application Example

The network stack of TSCH & 6TiSCH protocols has a large number of configurable parameters. The selection of these MAC-layer parameters has a significant impact on the network performance. Different parameters may be optimal for different deployments, depending on the application, the network topology, and the radio environment. Applications also impose constraints: the minimum and maximum acceptable parameter values. While some of these parameters can be found analytically, this is not always the case. As a result, experimental approach is often required [22]. A commonly used experimental strategy is called parameter sweep. It is a technique where a large number of simulations or pilot deployments are performed with different values of parameters in order exhaustively go through all potential parameter settings.

To put it more formally, let us consider a vector of parameters $p=(p_{1},\ldots ,p_{n})$ and define an optimization problem with constraints:$$\begin{array}{l} maximize\;(m_{1}\left(p\right),\ldots ,m_{k}\left(p\right))\\ \mathrm{subject}\;\mathrm{to}\;m_{1}\left(p\right)\geq c_{1},\ldots m_{k}\left(p\right)\geq c_{k}, \end{array}$$
where $m_{1},\ldots ,m_{k}$ are the performance metrics to optimize, and $c_{1},\ldots ,c_{k}$ the set of constraints to satisfy.

For the sake of an example, we show how to use TSCH-Sim for such a parameter selection via the parameter sweep technique. We assume that the data collection application (with packet period, packet size and other parameters specified in Table 2) needs to be deployed in a 300-node mesh network. For simplicity, we vary just a single parameter—the slotframe size sf: p=(sf), and look at the following metrics: PDR ($m_{1}$) and battery lifetime ($m_{2}$). More parameters and metrics can be easily added. We generate a random network topology with 300 nodes, and run the simulation ten times on this network, changing the slotframe size from 3 to 19 with a step of 2. To estimate the battery lifetime $m_{2}\left(p\right)$, we assume a 2600 mAh battery budget, common for off-the-shelf AA batteries. The results (Figure 12) show that longer slotframes are associated with monotonically increasing network lifetime, while also leading to monotonically decreasing PDR. Given a specific network optimization goal, the network designer can now use the results to select an appropriate combination of parameters. For example, if the goal is to optimize the battery lifetime while maintaining at least 95% PDR ($c_{1}=95\%$), then slotframe size of 9 is the best option (Figure 12).

## 6. Conclusions

This paper describes and evaluates TSCH-Sim, a new discrete event simulator for TSCH and 6TiSCH networks. Besides the core protocol, the simulator has support for multiple link connectivity models, mobility models, multiple TSCH schedulers including Orchestra, and the RPL routing protocol. The simulator comes with a module for summary metrics and a web interface for control and visualization. The evaluation shows that the simulator is an order of magnitude faster than existing alternatives without losing simulation accuracy, and that it can scale up to large networks, mostly due to a high abstraction level, improved link processing, and reduced log file sizes. In particular, we demonstrate faster-than-real-time simulations of a 10,000 node network on a regular PC.

## Figures and Tables

**Figure 1 sensors-20-05663-f001:**
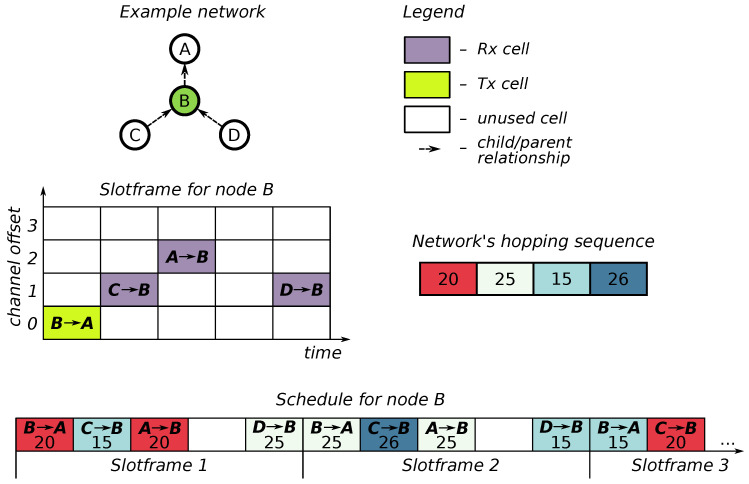
Overview of TSCH elements.

**Figure 2 sensors-20-05663-f002:**
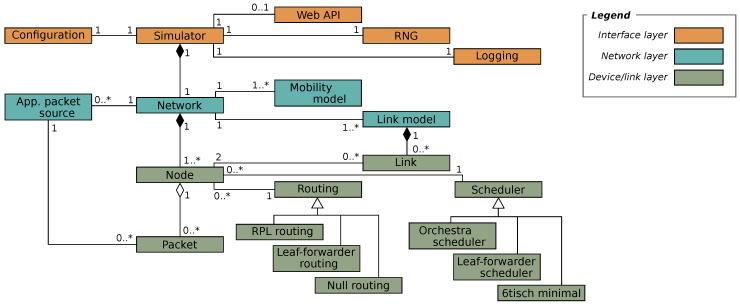
Class diagram of TSCH-Sim.

**Figure 3 sensors-20-05663-f003:**
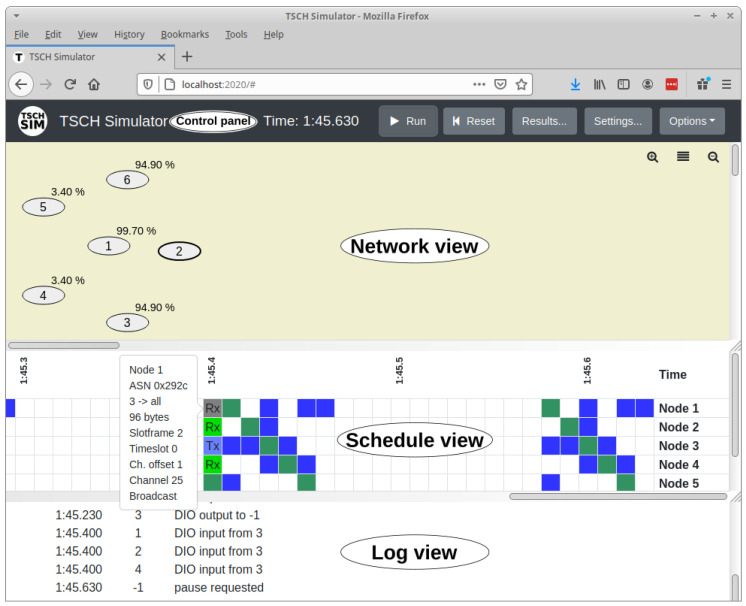
TSCH-Sim web interface.

**Figure 4 sensors-20-05663-f004:**
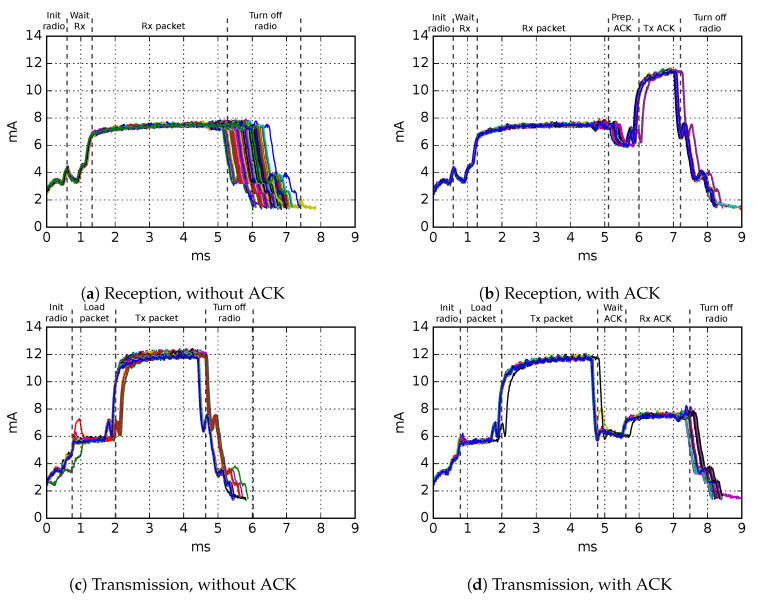
Current consumption measurements in TSCH timeslots. Data from 100 slots with 75-byte packets. The different colors denote measurements in different slots.

**Figure 5 sensors-20-05663-f005:**
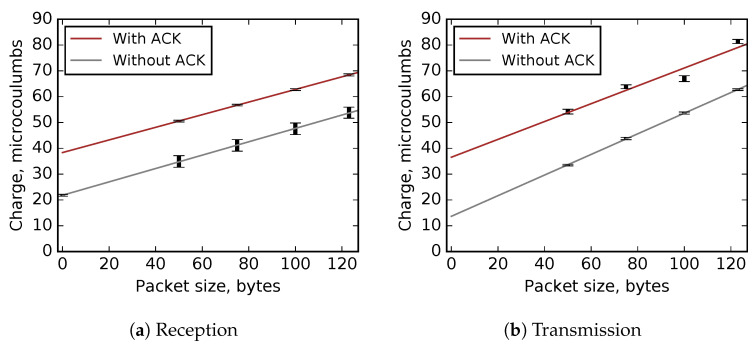
TSCH timeslot charge consumption profiles: measurements and the regression line. The bars show the minimal/maximal measurement range.

**Figure 6 sensors-20-05663-f006:**
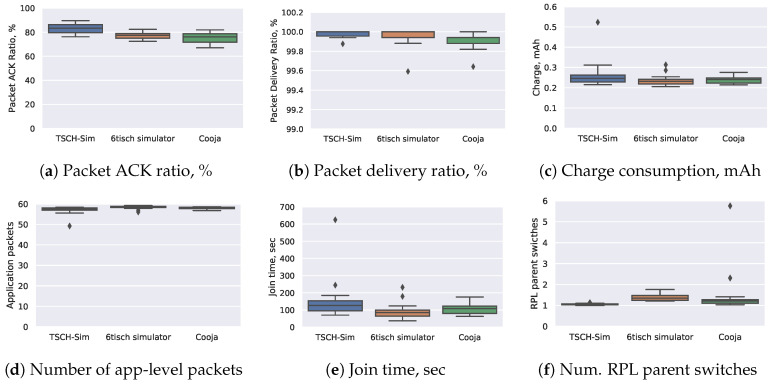
Comparison of results of 1 h long simulations in random 30-node networks.

**Figure 7 sensors-20-05663-f007:**
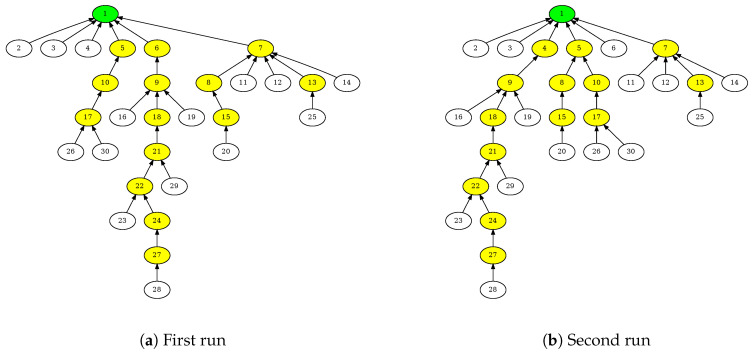
Different routing trees are created by the RPL protocol in different simulation runs.

**Figure 8 sensors-20-05663-f008:**
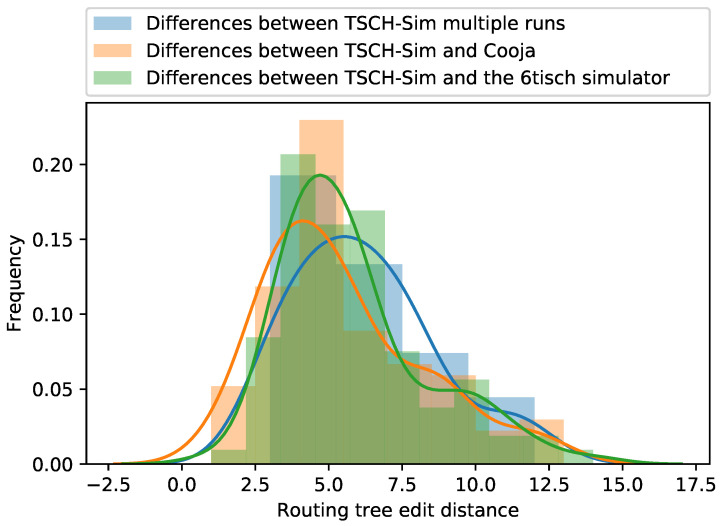
Similarity of the routing tree topologies.

**Figure 9 sensors-20-05663-f009:**
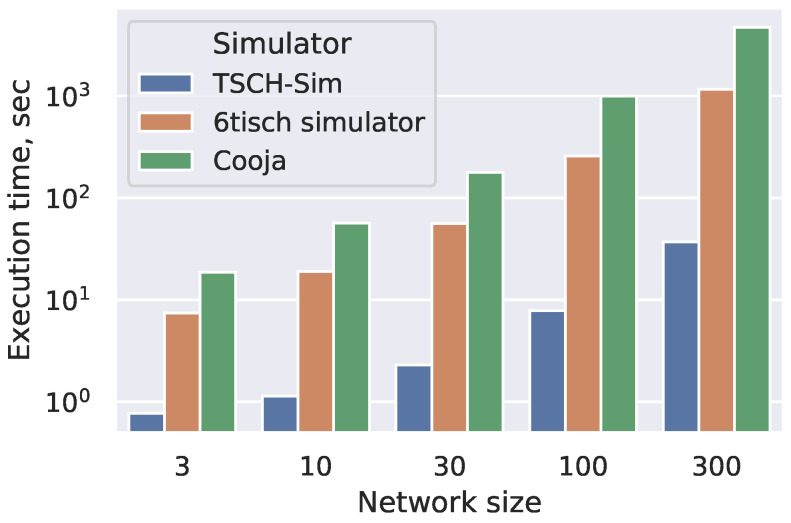
Simulation execution time depending on network size, 1 h simulations in random networks.

**Figure 10 sensors-20-05663-f010:**
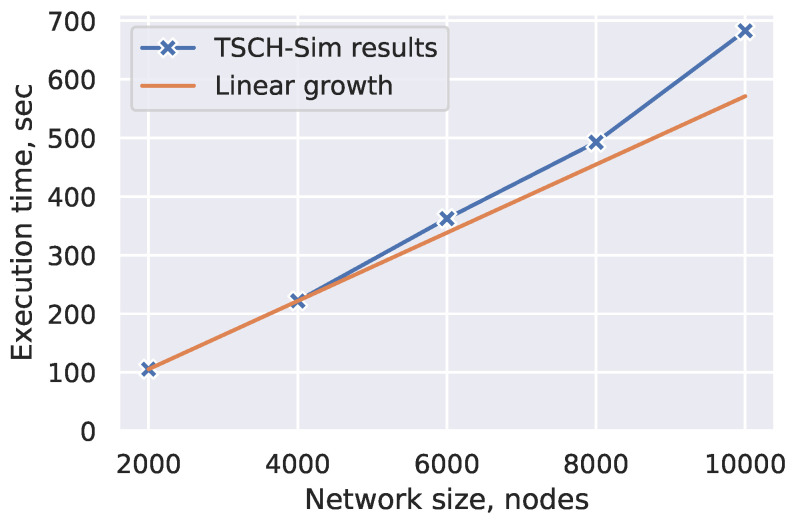
Simulation speed of 1 h long simulations in hierarchical networks.

**Figure 11 sensors-20-05663-f011:**
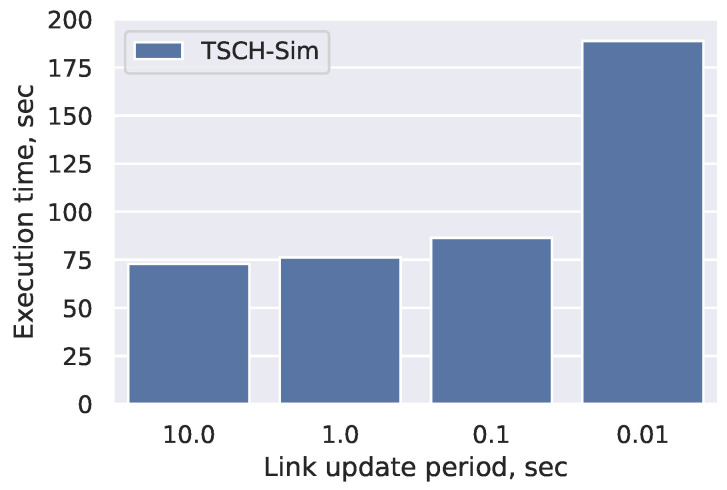
Impact of mobility on speed. 300 node mesh network; all nodes except the root are mobile.

**Figure 12 sensors-20-05663-f012:**
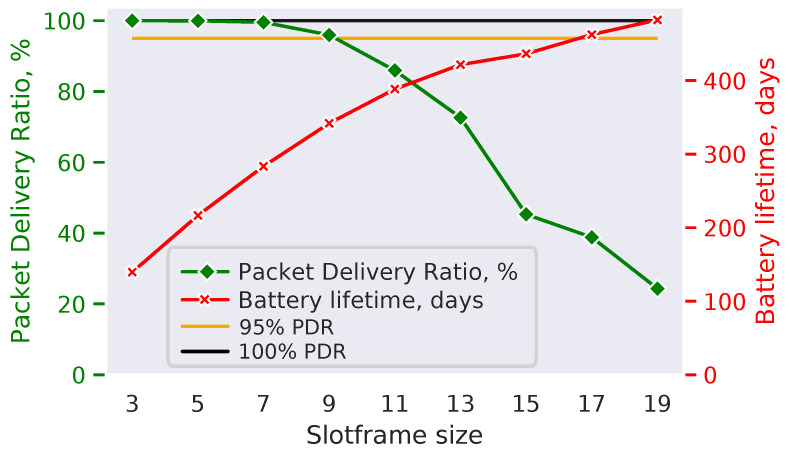
Parameter sweep results.

**Table 1 sensors-20-05663-t001:** TSCH simulator comparison.

	TSCH-Sim	6TiSCH	Cooja	OpenSim
Feature		Simulator		
IEEE TSCH	+	+	+	+
6TiSCH	partial	+	partial	+
RPL	+	+	+	+
Mobility	+	−	plugin	−
Trace-based simulation	+	+	plugin	−
Summary statistics	+	+	−	+
Charge consumption	+	+	−	+
C code execution	−	−	+	+
Hardware emulation	−	−	+	+
GUI	Web	Web	Java	Web and Python

**Table 2 sensors-20-05663-t002:** Experimental parameters.

Parameter	Value
Traffic type	data collection
Application packet interval	60 s
Queue size	8 packets
Number of channels	4 channels
Simulation duration	1 h
EB period	16 s
Keep-alive period	disabled
MAC retransmissions	7
Application packet size	100 bytes

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
