# Peer review of "TSCH-Sim: Scaling Up Simulations of TSCH and 6TiSCH Networks"

_sensors, 2020, doi:10.3390/s20195663_

Round 1

Reviewer 1 Report

1. It is compulsory to add more details about protocol frame structure and the timeslot occupied by the RPL protocol.
2. It is highly recommended to add examples about channel collisions modelling in TSCH-SIM, which can be reproduced in NS3, openWSN and OMNeT ++.
3. The calculation of charge consumption, which is mentioned in section 3.5 and shown in figure 5 is not described.
4. Authors do not provide a qualitative comparison with openWSN, OMNeT ++ for the TSCH protocol.
5. Figure 4 does not include any legends
6. Section 5 shows the dependence of the duration of the node operation on the size of the slot for a battery with a capacity of 2600mAh but there are no details about the intensity of sending messages

Author Response

See the attached document.

Reviewer 2 Report

This paper proposed a simulator for IEEE 802.15.4-2015 TSCH and 6TiSCH networks called TSCH-Sim with faster computational time by optimizing the link calculation and logging process. TSCH-Sim could serve experiments on large-scale networks. Besides providing basic TSCH simulator features, it also includes some useful features, such as mobility, reproducible experiment, and some sample experiments. The experiment shows that TSCH-Sim maintains better performance than prior tools.

The problem and contribution of this work are clearly identified and related. However, some points need to be considered:
- It is necessary to briefly explain the proposed method that makes TSCH-Sim have faster execution time than the prior tools into the Introduction and Conclusion sections.
- It is better to provide a design and implementation section that shows the Class diagram of the simulator and its explanation. Hence, it will help the user utilized it and also promote further customization.
- It also has some writing mistakes:
3.4.2 Unit Disk Graph -> Unit disk graph (sentence case). Including the same text inside this section.
3.4.5 Trace-based simulations -> Trace-based simulation (singular)

Reviewer 3 Report

The manuscript proposes a 802.15.4e TSCH simulator written in NodeJS JavaScript.

The paper is well written BUT:

  • MAJOR STUFF:
    • there IS ns3 code for tsch (https://github.com/EIT-ICT-RICH/ns-3-dev-TSCH)
    • there is also a simulator specifically done for the application the author is referencing (https://openwsn.atlassian.net/wiki/spaces/OW/overview)
    • I do not think it is a good idea to implement anything that needs to go fast over NodeJS, the fact that its speed is compared to other approaches that also use Java does not add any relevant information.
    • The fact that the link features are pre-computed and then updated periodically (ONLY in case of mobility) looks like a simplification too big. Even if it is justified by the fact that the other solutions have even more simple models does not mean anything compared to the per-packet link evaluation of ns3/omnet/openWsn. The same applies to propagation models, or their absence thereof.
  • MINOR STUFF
    • please reference something when you say that the other approaches are impractical to use or non-scalable (introduction).
    • how fast is the link update in case of mobility? At what update interval the simulation slows down to be comparable with the other approaches?
    • Please do not chastise ns3/omnet to be cumbersome to set up and then propose JSON files to edit manually as a mean to configure your approach. 
    • Are there any application models already available?
    • Figure 9 is very difficult to read in BW print.

Round 2

Reviewer 3 Report

The small corrections, that better specified the manuscript scope, enhanced the manuscript quality above the threshold for publication consideration.

However, I do not recommend immediate publications as (conditionally, please see below) there are still some small issues to resolve and the revision/response times are fast enough to justify this kind of granularity.

  • In the introduction, i do not "like" the use of the term "slow". One can say that one implementation focuses on computations that are not necessary in some scenarios, but not that it is "slow".
  • The author mention that it was impossible to perform more experiments due to time constraints implicit in the revision. I have asked the Editors to allow more time.
    • ONLY IF ALLOWED BY THE EDITORS please consider do a comparation campaign against OpenSim/OpenWSN.
  • About the implementing language discussion:
    1. if you had implemented  your approach using C/C++, would have it been "faster" ?
    2. just because something is popular it does not automatically mean that it is ontologically "right. Javascript, like Java, introduce layer after layer that can be metaphorically translated as "computational drag". Furthermore, to run it you still need NodeJS. 
    3. For the last point, please REMOVE any temporal reference to the "novelty" of Javascript (i.e. "modern").
  • The fact that some topologies are "unlikely" does not mean that they have to be left out.
    • ONLY IF ALLOWED BY THE EDITORS: a saturation study would have been nice if more time would have been give.

Author Response

> In the introduction, i do not "like" the use of the term "slow". One can say that one implementation focuses on computations that are not necessary in some scenarios, but not that it is "slow".

The term "slow" has been removed.

> The author mention that it was impossible to perform more experiments due to time constraints implicit in the revision. I have asked the Editors to allow more time.
ONLY IF ALLOWED BY THE EDITORS please consider do a comparation campaign against OpenSim/OpenWSN.

I have not done extra experiments due to very limited time (5 days) for this revision.

> About the implementing language discussion:
> if you had implemented your approach using C/C++, would have it been "faster" ?
> just because something is popular it does not automatically mean that it is ontologically "right. Javascript, like Java, introduce layer after layer that can be metaphorically translated as "computational drag". Furthermore, to run it you still need NodeJS.
> For the last point, please REMOVE any temporal reference to the "novelty" of Javascript (i.e. "modern").

I have remove the term "modern" from where it appeared (Introduction and "Core Functionality" Section).
A C or C++ implementation would indeed be faster if it was using the same algorithm. The prorgamming language benchmark shows that on the average, JS programs are around 3 times slower than the fastest language (typically C++) in the benchmark.
https://benchmarksgame-team.pages.debian.net/benchmarksgame/which-programs-are-fastest.html
https://benchmarksgame-team.pages.debian.net/benchmarksgame/fastest/node-gpp.html

> The fact that some topologies are "unlikely" does not mean that they have to be left out.
> ONLY IF ALLOWED BY THE EDITORS: a saturation study would have been nice if more time would have been give.

In Section "Scalability in Hierarchical Network" I have clarified: "The focus of these simulations is on sparse networks where the number of actual links is much lower than the number of potential connections.". I have not done extra experiments due to the limited time.